Evaluation of full-length nanopore 16S sequencing for detection of pathogens in microbial keratitis

Low Liying 1 2 lowliying@gmail.com
Fuentes-Utrilla Pablo 3
Hodson James 4
http://orcid.org/0000-0001-8205-7024 O’Neil John D. 5
Rossiter Amanda E. 6
Begum Ghazala 5 7
http://orcid.org/0000-0002-7884-8461 Suleiman Kusy 1
http://orcid.org/0000-0001-8491-3795 Murray Philip I. 1 2
Wallace Graham R. 1 2
Loman Nicholas J. 3 n.j.loman@bham.ac.uk
http://orcid.org/0000-0003-4627-3496 Rauz Saaeha 1 2
West Midlands Collaborative Ophthalmology Network for Clinical Effectiveness & Research by Trainees (WM CONCERT) 2
1 Academic Unit of Ophthalmology, Institute of Inflammation and Ageing, University of Birmingham , Birmingham, West Midlands , UK
2 Birmingham and Midland Eye Centre, Sandwell and West Birmingham Hospitals National Health Service (NHS) Trust , Birmingham, West Midlands , UK
3 MicrobesNG/School of Biosciences, University of Birmingham , Birmingham, West Midlands , UK
4 Queen Elizabeth Hospital, University Hospitals Birmingham NHS Foundation Trust , Birmingham, West Midlands , UK
5 Institute of Inflammation and Ageing, University of Birmingham , Birmingham, West Midlands , UK
6 Institute of Microbiology and Infection, University of Birmingham , Birmingham, West Midlands , UK
7 National Institute for Health Research Surgical Reconstruction and Microbiology Research Centre , Birmingham , UK
Khiabanian Hossein
Electronic publication date: 2021 Feb 15
Publication date: 2021
Volume: 9
Electronic Location ID: e10778
Received 2020 May 4; Accepted 2020 Dec 22
Copyright: © 2021 Low et al.
Copyright year: 2021
Copyright holder: Low et al.
License: This is an open access article distributed under the terms of the Creative Commons Attribution License, which permits unrestricted use, distribution, reproduction and adaptation in any medium and for any purpose provided that it is properly attributed. For attribution, the original author(s), title, publication source (PeerJ) and either DOI or URL of the article must be cited.
License URL: https://creativecommons.org/licenses/by/4.0/

Keywords: Nanopore sequencing, Eye infection, Microbial keratitis, Full length 16S rRNA sequencing, Cornea infection, Eye swab, 16S bioinformatics, Corneal infection, Ophthalmology, Molecular diagnostics

Funding: Royal College of Ophthalmologists and Fight for Sight Ophthalmology Trainee Research Network Award 24CO3 National Institute for Health Research (NIHR) Surgical Reconstruction & Microbiology Research Centre (SRMRC)/Royal Centre for Defence Medicine, Ministry of Defence (UK) RCDM–ADMST0003 Fight for Sight Clinical Research Fellowship 1840/41 This study has been funded through the Royal College of Ophthalmologists and Fight for Sight Ophthalmology Trainee Research Network Award (Ref. 24CO3), and the National Institute for Health Research (NIHR) Surgical Reconstruction and Microbiology Research Centre (SRMRC)/Royal Centre for Defence Medicine, Ministry of Defence (UK) grant (Ref. RCDM—ADMST0003: ‘Rapid Diagnosis of Corneal Infections’). Dr Liying Low is funded by a Fight for Sight Clinical Research Fellowship (Ref. 1840/41). The funders had no role in study design, data collection and analysis, decision to publish, or preparation of the manuscript.

==============================
Background

Microbial keratitis is a leading cause of preventable blindness worldwide. Conventional sampling and culture techniques are time-consuming, with over 40% of cases being culture-negative. Nanopore sequencing technology is portable and capable of generating long sequencing reads in real-time. The aim of this study is to evaluate the potential of nanopore sequencing directly from clinical samples for the diagnosis of bacterial microbial keratitis.

Methods

Using full-length 16S rRNA amplicon sequences from a defined mock microbial community, we evaluated and benchmarked our bioinformatics analysis pipeline for taxonomic assignment on three different 16S rRNA databases (NCBI 16S RefSeq, RDP and SILVA) with clustering at 97%, 99% and 100% similarities. Next, we optimised the sample collection using an ex vivo porcine model of microbial keratitis to compare DNA recovery rates of 12 different collection methods: 21-gauge needle, PTFE membrane (4 mm and 6 mm), Isohelix™ SK-2S, Sugi® Eyespear, Cotton, Rayon, Dryswab™, Hydraflock®, Albumin-coated, Purflock®, Purfoam and Polyester swabs. As a proof-of-concept study, we then used the sampling technique that provided the highest DNA recovery, along with the optimised bioinformatics pipeline, to prospectively collected samples from patients with suspected microbial keratitis. The resulting nanopore sequencing results were then compared to standard microbiology culture methods.

Results

We found that applying alignment filtering to nanopore sequencing reads and aligning to the NCBI 16S RefSeq database at 100% similarity provided the most accurate bacterial taxa assignment. DNA concentration recovery rates differed significantly between the collection methods (p < 0.001), with the Sugi® Eyespear swab providing the highest mean rank of DNA concentration. Then, applying the optimised collection method and bioinformatics pipeline directly to samples from two patients with suspected microbial keratitis, sequencing results from Patient A were in agreement with culture results, whilst Patient B, with negative culture results and previous antibiotic use, showed agreement between nanopore and Illumina Miseq sequencing results.

Conclusion

We have optimised collection methods and demonstrated a novel workflow for identification of bacterial microbial keratitis using full-length 16S nanopore sequencing.

Introduction

Microbial keratitis is a leading cause of preventable blindness worldwide and is the most common cause of acute medical ophthalmology admission (Centers for Disease Control and Prevention, 2014). Bacterial keratitis accounts for the majority of the microbial keratitis cases in the Western hemisphere, with a preponderance for Gram-positive organisms such as Staphylococcus aureus and Streptococcus pneumoniae, and to a lesser extent Gram-negative organisms such as Pseudomonas aeruginosa and Klebsiella pneumoniae (Ting et al., 2018; Tan et al., 2017; Ibrahim, Boase & Cree, 2009; Lichtinger et al., 2012). Prognosis is dependent on early identification of the causative organism and initiation of appropriate treatment, including antibiotic therapy (Austin, Lietman & Rose-Nussbaumer, 2017). Conventional sampling and culture technique is time-consuming, with culture results taking 48 hours and antimicrobial sensitivity results taking up to 5 days (Maurer et al., 2017). Approximately 40% of clinically suspected microbial keratitis cases are culture-negative (Sugita et al., 2013; Tananuvat et al., 2012), leading to the widespread initial implementation of empirical broad-spectrum antimicrobial therapeutic protocols, increasing the risk of antimicrobial resistance and poorer patient outcomes (Goldstein, Kowalski & Gordon, 1999), particularly in patients who do not have ready access to diagnostic laboratories, such as those in poor and low income countries and deployed military personnel (Musa et al., 2010).

Culture-independent molecular techniques such as polymerase chain reaction (PCR) has been utilised in the diagnosis of ocular infections (Kim et al., 2008), however, this requires a priori knowledge of the likely pathogenic micro-organism to determine which specific primer sets to use and is limited by the number of species that can be detected simultaneously in a single PCR assay (Bispo et al., 2018; Ung et al., 2020). Newer, high-throughput sequencing approaches can be broadly classified into two major options: targeted amplicon sequencing (selective amplification of the specific genetic region of interest such as 16S rRNA in bacteria or 18S rRNA in fungi) and metagenomic sequencing (untargeted amplification of all genomic DNA) (Ung et al., 2020). Untargeted metagenomic sequencing allows for the discovery of unexpected, novel pathogens, such as Vittaforma corneae in infectious conjunctivitis (Lalitha et al., 2019) and Torque teno virus in culture-negative endophthalmitis (Lee et al., 2015). Deep metagenomic sequencing techniques have enabled phylogenetic analysis of the temporal and geographic origin of ocular infection (Doan et al., 2016b; Kirstahler et al., 2018).

Several challenges have hindered the adoption of sequencing technologies in routine clinical practice for microbial keratitis, including sample collection methods that are frequently ineffective (Kaye et al., 2003) compounded by the low abundance of pathogens (Ung et al., 2020), high contamination from background host DNA or laboratory reagents (Gu, Miller & Chiu, 2019), lack of standardisation in sequencing and bioinformatics processing methods (Chiu & Miller, 2019). Sampling and DNA extraction methodology significantly impact upon the downstream sequencing data in samples with low biomass (Douglas et al., 2020; Sui et al., 2020). Microbial cells within the sample must be sufficiently lysed to liberate its DNA content, and this is particularly challenging for thick-walled microorganisms whereby mechanical disruption method such as bead-beating in conjunction with heat, chemical or enzymatic treatment is required in the DNA extraction protocol (De Boer et al., 2010; Ojo-Okunola et al., 2020). Clinical samples have high host DNA content, usually constituting more than 90% of the sequences, with relatively low abundance of pathogen DNA, and therefore requiring greater depth of sequencing (Ung et al., 2020; Lalitha et al., 2019). The prohibitively high running costs of large sequencing platforms mean that they are only available in select centres, and samples are generally pooled for batch processing, resulting in delayed turnaround time. Efforts have been directed to reduce the amount of host DNA present in clinical samples, especially for samples that can be obtained in abundance, such as saliva, sputum, and bronchoalveolar lavage and blood (Charalampous et al., 2019; Marotz et al., 2018; Feehery et al., 2013). However these techniques are still not compatible with a generic method that can be applied across all the different types of clinical samples.

Marker genes using conserved, housekeeping regions of the genome interspersed with variable regions have been utilised to infer phylogenetic links and microbial taxonomy (Clarridge, 2004; Raina et al., 2019). The universal 16S ribosomal RNA (rRNA) gene sequence in bacteria, which is approximately 1,550 base pairs long, composed of a highly conserved region interspersed with nine variable regions (V1-9), is the most commonly used marker gene for assessing bacterial profiles (Clarridge, 2004; Akram et al., 2017; Achtman & Wagner, 2008). Amplification of the 16S rRNA region by PCR reduces the background host contamination and by only sequencing a smaller part of the genome, this drastically reduces the sequencing depth requirements and cost. The vast majority of clinical studies have only sequenced part of the 16S gene, ranging from the single variable region of V4, V6 to three variable regions V1-3 or V3-5, because the widely used Illumina sequencing platform is only capable of producing short reads of less than 500 bases (Johnson et al., 2019; Holm et al., 2019). This short-read sequencing further compounds the taxonomic resolution of 16S rRNA sequencing, which is typically limited to genus-level resolution (Achtman & Wagner, 2008). Choice of the hypervariable region affects the taxonomic assignment, for example the V4 hypervariable region providing better whole bacterial diversity in human gut microbiome studies whereas the V1-V2 hypervariable region is more specific for skin microbiota profiling (Santos et al., 2020). Several studies have shown that full-length 16S rRNA reads provides better taxonomic resolution compared to reads that only target a certain region of the 16S rRNA gene (Winand et al., 2020; Nygaard et al., 2020). The choice of 16S reference database also impacts upon the taxonomic assignment (Nygaard et al., 2020; Park & Won, 2018; Szabó et al., 2016). For example, the expanded Human Oral Microbiome Database contains references to microbes specifically from the aerodigestive tract whilst the National Center for Biotechnology Information (NCBI) 16S RefSeq, the Ribosomal Database Project (RDP) and the SILVA rRNA databases include taxa from all sources (human and non-human hosts) and the environment (RefSeq, 2020; Cole et al., 2014; Quast et al., 2013; Escapa et al., 2018).

With the introduction of long-read sequencing technologies, such as the Pacific Biosciences (PacBio) and Oxford Nanopore Technologies (ONT) sequencing platforms, we are now able to sequence in real-time the full-length of the 16S rRNA gene (V1–V9 regions) (Johnson et al., 2019; Winand et al., 2020). Major advantages of using nanopore sequencing include portability (pocket-sized ONT MinION sequencer vs large PacBio sequencer), relatively low cost (~£140 per sample by multiplexing 12 samples per run and £820 for the ONT MinION vs ~£280 per sample and £390,000 for the PacBio Sequel II System) and rapid workflow (less than an hour from sample preparation to data analysis with nanopore sequencing vs 3.5 hours with PacBio) (De Maio et al., 2019; 16S Sequencing and Analysis, 2020; Ashton et al., 2015; PacBio Sequel Systems, 2020; Workflow-PacBio, 2020). Nanopore sequencing has been utilised for on-the-field, real-time genomic surveillance of Ebola (Quick et al., 2016) and Zika (Quick et al., 2017; Faria et al., 2017) viruses. In addition, nanopore sequencing has also been used for pathogen detection in proof-of-concept clinical studies of sepsis (Leggett et al., 2020), lower respiratory tract (Charalampous et al., 2019; Yang et al., 2019), urinary tract (Schmidt et al., 2017) and prosthetic joint infections (Sanderson et al., 2018). However, there is relatively high error rates in nanopore sequencing (~95% raw read accuracy for nanopore) (Rang, Kloosterman & De Ridder, 2018), which affects the discriminatory power of 16S rRNA gene for species level classification (Winand et al., 2020). Strategies to reduce nanopore sequencing error rate are constantly evolving with improvements to the pore chemistry and basecalling software (Rang, Kloosterman & De Ridder, 2018).

The aim of this study was to evaluate the potential of full-length 16S nanopore sequencing directly from clinical samples of bacterial microbial keratitis, focussing on the optimisation of the DNA extraction and bioinformatics pipeline to make 16S nanopore sequencing feasible. Firstly, using a defined mock microbial community, we evaluated and benchmarked our bioinformatics analysis pipeline for taxonomic assignment. Then, we optimised the sample collection using an ex vivo porcine model of microbial keratitis. Next, we performed a proof-of-concept study in which we applied the sampling technique that provided the highest DNA recovery and the optimised bioinformatics pipeline to prospectively collected samples from patients with suspected microbial keratitis, comparing the nanopore sequencing results to standard microbiology culture methods (Fig. 1).

Figure 1 Overview of study workflow.

Study workflow starting from in silico studies for bioinformatics benchmarking on defined mock community, ex vivo study for optimising eye swab/collection methods and proof-of-concept clinical study on patients with microbial keratitis. Abbreviations: Staphylococcus aureus (S. aureus); Klebsiella pneumoniae (K. pneumoniae); Enterococcus avium (E. avium); National Center for Biotechnology Information (NCBI); Ribosomal Database Project (RDP); quantitative polymerase chain reaction (qPCR).

Materials and Methods

Mock bacterial community

Bacterial species representative of the spectrum of microbial keratitis causative Gram-positive and Gram-negative pathogens (Staphylococcus aureus, Klebsiella pneumoniae and Enterococcus avium (previously classified as group D Streptococcus)) were used to define the mock bacterial community. Klebsiella pneumoniae and Staphylococcus aureus were grown overnight at 37 °C in lysogeny broth (LB) whilst Staphylococcus aureus was grown in Brain Heart Infusion (BHI) broth. Negative controls consisted of the LB and BHI broth without any inoculum, respectively. Overnight cultures were diluted into fresh medium to an optical density 600 nm (OD600) of 0.05 and incubated at 37 °C with aeration. The culture OD600 was measured every 30 min for 5 h using spectrophotometer (Ultrospec™ 2100 pro, Amersham Biosciences, UK). For enumeration of bacteria, cultures were plated onto agar plates (Enterococcus avium, blood agar; Staphylococcus aureus and Klebsiella pneumoniae, LB agar respectively) and incubated overnight at 37 °C. The colony-forming units (CFUs) for each species were enumerated the following day. Bacterial growth curves were plotted and the mid-exponential growth phases of the samples were taken. The mock bacterial community consisted of 1 × 105 CFU/ml each of Klebsiella pneumoniae, Enterococcus avium and Staphylococcus aureus.

Ex vivo porcine model of microbial keratitis

Freshly enucleated porcine eyes (Sus scrofa domestica) were obtained as a by-product of the meat industry and transported to the laboratory under storage at 4 °C. Each eye was disinfected with Povidone-iodine 10% w/w for 1 min, followed by two rinses of sterile 0.9% Sodium Chloride for 1 min, and placed in an individual chamber of a sterile 6-well culture plate (Sigma–Aldrich, Merck KGaA, Darmstadt, Germany). Using stereoscopic surgical loupes, a 4mm trephine punch (Acu-Punch®, Acuderm, Fort Lauderdale, FL, USA) was used to create a single central anterior stromal corneal lesion (with debridement of the central 4 mm). The area was sampled with the respective swabs/collection methods pre-inoculation, as a background control. Each eye was then inoculated with 20 µL of 1 × 105 CFU/ml each of the mock community (Enterococcus avium, Staphylococcus aureus and Klebsiella pneumoniae). The amount of inoculum was determined according to the estimated density of bacteria encountered in vivo in clinical infections (Kaye et al., 2003). The area was re-sampled with the respective collection methods after 30 min to prevent bacterial overgrowth, and placed immediately into a ZR BashingBead™ Lysis Tube containing 750 µl of DNA Shield™ (Zymo Research, Irvine, CA, USA) and stored at −80 °C until DNA extraction.

Collection method/swabs

The different collection methods evaluated are Isohelix™ SK-2S, Sugi® Eyespear, Cotton, Rayon, Dryswab™, Hydraflock, Albumin-coated, Purflock, Purfoam, Polyester swabs, 21-gauge needle, and Polytetrafluoroethylene (PTFE) membranes (Table 1). Sampling order was varied. A total of 60 ex-vivo porcine eye models were used. Each sampling replicate was performed on a separate eye. Swabs were not pre-moistened, as a previous pilot study that we have conducted showed that dry swabs provided higher DNA yield, compared to pre-moistened swabs (Table S1).

Table 1 Summary of the collection methods used.

Name	Manufacturer	Composition of bud	Shaft	Catalogue number	
Isohelix™ SK-2S	Isohelix, Kent, UK	Viscose rayon	–	SK-2S	
Sugi® Eyespear	Kettenbach GmbH and Co. KG, Escehnburg, Germany	Cotton and cellulose	–	30901	
MW1021 Dryswab™ Rayon	MWE Medical Wire, Corsham, UK	Rayon	Plastic	MW1021	
MW1041 Cotton	MWE Medical Wire, Corsham, UK	Cotton	Wood	MW1041	
MW1021D Dryswab™ Polyester	MWE Medical Wire, Corsham, UK	Polyester	Plastic	MW1021D	
MW821 Dryswab™ Flock	MWE Medical Wire, Corsham, UK	Purflock®	Plastic	MW831	
MW100 Fine tip Dryswab™ Rayon	MWE Medical Wire, Corsham, UK	Rayon	Plastic	MW100	
MW130 Hospiswab™ Albumin	MWE Medical Wire, Corsham, UK	Albumin coated	Wood	MW130	
MW840 Hydraflock® Plastic	MWE Medical Wire, Corsham, UK	Hydraflock®	Plastic	MW840	
MW946 Sigma Swab® Purfoam	MWE Medical Wire, Corsham, UK	Purfoam	Plastic	MW946	
BD Needle 21G	Becton-Dickinson (BD) & Co, New Jersey, USA	Stainless steel	–	BD301155	
Biopore® PTFE 6 mm	Merck & Co, New Jersey, USA	PTFE	–	BGCM00010	
Biopore® PTFE 4 mm	Merck & Co, New Jersey, USA	PTFE	–	BGCM00010	
Note:

Becton-Dickinson (BD); Medical Wire and Equipment (MWE); Polytetrafluoroethylene (PTFE).

DNA extraction

DNA from the mock community, conjunctival swabs and negative controls were extracted using ZymoBIOMICS DNA Miniprep kit (Zymo Research, Irvine, CA, USA) according to the manufacturer’s instructions. Negative control DNA extraction was performed on reagents without a DNA template and also on sterile 0.9% Sodium Chloride that was used to clean the porcine eyes, whilst positive control DNA extraction was performed on 20 µL of 1 × 105 CFU/ml each of Enterococcus avium, Staphylococcus aureus and Klebsiella pneumoniae inserted directly into the lysis tube containing 750 µl of DNA Shield™.

DNA concentration was determined fluorometrically using a Qubit dsDNA high-sensitivity assay (Thermo Fisher Scientific, Waltham, MA, USA).

16S qPCR for 16S and beta-actin

16S qRT-PCR for 16S and beta-actin genes was done to compare the host to microbial DNA ratio across the different sampling methods.

The primers used to amplify the 16S rRNA gene were: forward primer 341F 5′-CCTACGGGAGGCAGCAG-3′, and reverse primer 534R 5′-ATTACCGCGGCTGCTGGCA-3′. These primers are complementary to the conserved regions in the 16S rRNA gene, nucleotide positions 290–484 in Escherichia coli, producing a fragment of 195 bp.

The primers used to amplify the beta-actin Sus scrofa gene were: Forward primer 5′-CCAAGCCTGGACTACCTCCT-3′, and reverse primer 5′-AAACCTGGAGAGGTTCACCG-3′. These primers were complementary to the Sus scrofa actin beta (ACTB) transcript RNA gene, spanning the nucleotide positions 1,348–1,535, producing an amplicon length of 188 bp.

Primers were synthesized by Eurofins Genomics (Ebersberg, Germany). The final PCR mix contained 0.4 μl each of forward and reverse primers (total concentration of 0.4 nmol each), 5 μl of SYBR Green Master Mix (TaKaRa BioTech Corporation, Dalian, China), 3.2 μl of UltraPure™ DNase/RNase-Free distilled water (ThermoFisher Scientific, Waltham, MA, USA) and 1 μl of unamplified genomic DNA, giving a final reaction volume of 10 μl. All samples were performed in triplicate. qPCR was performed using the Roche LightCycler®480 Instrument (Roche Diagnostics, Meyland, France) on the following programme: 1 cycle of 30 s at 95 °C, 40 cycles of 10 s at 95 °C, 30 s at 59 °C and 20 s at 78 °C.

16S gene copy number quantification

The genomic 16S copy number was quantified using Eubacteria 16S Ribosomal gene genesig® Standard kit (Primerdesign™ Ltd, Camberley, UK). The final PCR mix contained 10 μl of PrecisionPLUS 2X qPCR Master Mix, 1 μl of Eubacteria/probe mix and 5 μl of unamplified genomic DNA, and 4 μl of RNase/DNase free water giving a final reaction volume of 15 μl. Samples were divided and performed in triplicates. For the standard curve, the positive control template (16S Eubacteria) was serially diluted 10-fold to obtain copy numbers ranging from 2 × 105 to 2 copy/μL. Quantitative PCR was performed with the Roche LightCycler®480 Instrument (Roche Diagnostics, Meyland, France) on the following programme: 1 cycle of 2 min at 95 °C, 50 cycles of 10 s at 95 °C and 1 min at 60 °C. The CT number of experimental samples were interpolated against the standard curve to calculate the corresponding 16S copy number.

Nanopore 16S barcoding, sequencing and bioinformatics

The full-length 16S rRNA genes (16S sequencing primers: 27F-AGAGTTTGATCMTGGCTCAG; 1492R-CGGTTACCTTGTTACGACTT) were amplified and samples with amplicons above 1nM were subsequently sequenced on GridION (ONT, Oxford, UK) using the R9 flow cell (FLO-MIN106; ONT, Oxford, UK), as per manufacturer’s protocol. Raw sequence reads were basecalled using ONT’s MinKNOW software Guppy v.3.2.4 with the R9.4 high accuracy model. Raw reads were demultiplexed using qcat version 1.0.1. Sequence summary and read length histograms were generated using NanoPlot version 1.30.1. 16S rRNA databases were obtained from NCBI 16S RefSeq (RefSeq, 2020) (Nucleotide search details: 33,175 (BioProject) or 33,317 (BioProject)); the RDP (Cole et al., 2014) release 11, update 5; and the SILVA rRNA database project (Cole et al., 2014) version 132 repositories respectively. Differences in the characteristics of the three databases are summarised in Table S2. Reference sequences from the 16S rRNA databases were then clustered into 97%, 99% and 100% similarity thresholds using heuristic clustering method (greedy incremental clustering algorithm) on CD-HIT (Fu et al., 2012) version 4.8.1 (commands: -c 0.97, 0.99, 1.0, -M 62900, -d 250). We used Minimap2 (Li, 2018) version 2.12-r87 (commands: -K 100M, -ax map-ont) to align the demultiplexed reads to the respective 16S databases and processed the resulting files using the Sequence Alignment/Map (SAM)tools (Li et al., 2009) version 1.9 (commands: samtools view -b -F 2308 (to remove unmapped, non-primary and supplementary reads), samtools sort, samtools index, samtools idxstats). Bioinformatics scripts and the FASTQ files are available at DOI 10.6084/m9.figshare.13213898.v1.

Clinical sample collection from patients presenting with microbial keratitis

Patients (n = 2) presenting to the Birmingham and Midland Eye Centre, Birmingham, West Midlands, UK with suspected microbial keratitis affecting one eye were recruited to the study. Informed written consent was obtained. The research followed the tenets of the Declaration of Helsinki and was approved by the Health Research Authority Ethics Committee (Rapid Diagnosis of Ocular Infections (RADAR); Reference: 11/EM/0274).

Standard clinical microbiology corneal scrape culture results were compared with nanopore sequencing. For each patient, corneal scrapes were taken for standard clinical microbiology as per routine clinical practice (Lin et al., 2019). Corneal scrapes were taken at the base and edge of the lesion, inoculated onto blood, chocolate and Sabaraud’s agar plates, and processed at the local clinical microbiology laboratory (The agar plates were incubated for 5 days—blood and chocolate (6% CO2 at 37 °C); Sabourand’s (air at 30 °C)). The collection method with the highest DNA recovery in the ex vivo porcine study was used to collect corneal, conjunctival and negative control samples from the patients. Swabs were taken from the affected cornea and unaffected contralateral conjunctiva, together with a negative control ‘air swab’ of the examination room at the time point of participant sampling, to exclude contamination. Swabs were placed immediately into a ZR BashingBead™ Lysis Tube containing 750 ul of DNA Shield™ (Zymo Research, Irvine, CA, USA) and stored at −80 °C until DNA extraction, as described above. Sequence data were compared with the corresponding microbiology culture results of the hospital corneal scrapes. In cases where there was no concordance between culture and nanopore sequencing results, Illumina MiSeq 16S rRNA V4 sequencing was performed.

Miseq 16S sequencing and bioinformatics

The V4 variable region of the 16S rRNA gene was amplified from DNA extracts using primer sets (515F: GTGCCAGCMGCCGCGGTAA; and 806R: GGACTACHVGGGTWTCTAAT) and sequenced on the Illumina MiSeq platform (Kozich et al., 2013). Raw data were filtered and analysed with Mothur and QIIME software packages, as previously described (Kozich et al., 2013; Caporaso et al., 2010).

Statistical analysis

Comparisons of DNA concentrations and CT across the collection materials were performed using Kruskal-Wallis tests. The mean ranks for the DNA concentrations were then used to sort the collection material in order, and data were summarised using medians and ranges. All analyses were performed using IBM SPSS 22 (IBM Corp. Armonk, NY, USA), with p < 0.05 deemed to be indicative of statistical significance throughout.

Results

Evaluation of bioinformatics analysis pipeline on defined mock microbial community

To evaluate the different methods of performing taxonomic assignment for the most reliable bacterial taxonomic assignment on 16S rRNA nanopore sequencing reads, we compared the effects of removal of unmapped, non-primary and supplementary reads on the taxonomic identification of the mock community (Table 2; Figs. S1 and S2). Removal of unmapped, non-primary and supplementary reads provided a more accurate taxonomic assignment of the mock community.

Table 2 Comparing the total mapped reads and taxa identification of reads that have not been filtered vs. reads that have been filtered to remove unmapped, non-primary and supplementary reads.

Without removal of unmapped, non-primary and supplementary reads	With removal of unmapped, non-primary and supplementary reads	
Total alignments	705,919		Total alignments	117,827		
Taxa	#Mapped	Relative abundance	Taxa	#Mapped	Relative abundance	
Enterococcus avium*	204,171	28.92	Enterococcus avium*	64,172	54.46	
Klebsiella pneumoniae*	183,274	25.96	Klebsiella pneumoniae*	32,248	27.37	
Enterococcus pseudoavium	81,810	11.59	Staphylococcus aureus*	14,772	12.54	
Staphylococcus aureus*	72,691	10.30	Enterococcus malodoratus	913	0.77	
Enterococcus malodoratus	46,019	6.52	Enterococcus pseudoavium	868	0.74	
Enterococcus devriesei	21,978	3.11	Enterococcus viikkiensis	535	0.45	
Enterococcus viikkiensis	15,614	2.21	Enterococcus hirae	410	0.35	
Staphylococcus simiae	13,980	1.98	Enterococcus devriesei	294	0.25	
Enterococcus gilvus	7,862	1.11	Enterococcus durans	203	0.17	
Enterococcus hirae	6,380	0.90	Enterobacter cloacae	170	0.14	
Taxa			Taxa			
Enterococcus	409,683	58.04	Enterococcus	68,665	58.28	
Klebsiella	187,581	26.57	Klebsiella	32,482	27.57	
Staphylococcus	90,775	12.86	Staphylococcus	15,184	12.89	
Serratia	3,078	0.44	Enterobacter	214	0.18	
Enterobacter	3,011	0.43	Serratia	165	0.14	
Pectobacterium	1,238	0.18	Pectobacterium	124	0.11	
Streptococcus	763	0.11	Streptococcus	83	0.07	
Citrobacter	653	0.09	Citrobacter	76	0.06	
Bacillus	601	0.09	Bacillus	73	0.06	
Lactobacillus	579	0.08	Lactobacillus	51	0.04	
Note:

* Mock microbial community in the positive control sample—Enterococcus avium, Staphylococcus aureus and Klebsiella pneumoniae Reads were mapped against the unclustered NCBI 16S database Relative abundance is defined as the number of reads mapped to the taxa divided by the total number of mapped reads.

Clustering of reference databases by a threshold of sequence similarity may reduce search time and make mapping faster (Li, Jaroszewski & Godzik, 2002). To determine the effects of clustering, the reference sequences of three different 16S databases were reviewed (the NCBI Reference Sequence, NCBI 16S RefSeq; the RDP; and the SILVA ribosomal RNA gene database) (Table 3). We found that for all three databases, clustering at a threshold of 99% and above provided more accurate operational taxonomic unit assignment (OTU) compared to clustering at 97% similarity. The NCBI 16S RefSeq database with reference sequences clustered at 100% similarity provided the most accurate OTU assignment, with more than 95% of the reads corresponding to the mock community. Based on these results, we chose to align our sequencing reads to the NCBI 16S RefSeq database at 100% similarity (reference database sequences clustered at 100% similarity) and removed any unmapped, non-primary or secondary alignments.

Table 3 Comparing the effects of using NCBI 16S, RDP and SILVA databases at different levels of clustering (97%, 99% and 100% similarity).

NCBI 16S RefSeq	RDP	SILVA	
Number of reads	Cluster	#Mapped	Relative abundance	Number of reads	Cluster	#Mapped	Relative abundance	Number of reads	Cluster	#Mapped	Relative abundance	
482,900	97% similarity			483,062	97% similarity			481,985	97% similarity			
	Enterococcus hirae	183,027	37.90		Enterobacter soli	173,164	35.85		Klebsiella quasipneumoniae	114,637	23.78	
	Enterobacter soli	172,739	35.77		Enterococcus faecium	123,393	25.54		Enterococcus faecium	109,825	22.79	
	Staphylococcus aureus*	99,582	20.62		Staphylococcus aureus*	99,695	20.64		Staphylococcus aureus*	99,153	20.57	
	Salmonella enterica	7,266	1.50		Enterococcus asini	21,726	4.50		Klebsiella pneumoniae*	54,508	11.31	
	Enterococcus casseliflavus	2,761	0.57		Enterococcus phoeniculicola	19,122	3.96		Enterococcus casseliflavus	34,290	7.11	
483,211	99% similarity			483,374	99% similarity			482,392	99% similarity			
	Klebsiella pneumoniae*	184,328	38.15		Klebsiella pneumoniae*	187,465	38.78		Klebsiella pneumoniae*	164,929	34.19	
	Enterococcus avium*	173,319	35.87		Enterococcus avium*	182,655	37.79		Enterococcus avium*	109,569	22.71	
	Staphylococcus aureus*	97,455	20.17		Staphylococcus aureus*	96,923	20.05		Staphylococcus aureus*	98,477	20.41	
	Enterococcus hirae	4,975	1.03		Enterococcus malodoratus	1,281	0.27		Enterococcus devriesei	39,961	8.28	
	Enterococcus malodoratus	4,401	0.91		Enterobacter cloacae	1,235	0.26		Enterococcus raffinosus	24,276	5.03	
483,382	100% similarity			483,432	100% similarity			482,599	100% similarity			
	Klebsiella pneumoniae*	187,745	38.84		Klebsiella pneumoniae*	179,200	37.07		Klebsiella pneumoniae*	184,252	38.18	
	Enterococcus avium*	175,497	36.31		Enterococcus avium*	176,822	36.58		Enterococcus avium*	112,163	23.24	
	Staphylococcus aureus*	97,505	20.17		Staphylococcus argenteus	90,126	18.64		Staphylococcus aureus*	98,461	20.40	
	Enterococcus pseudoavium	3,758	0.78		Staphylococcus aureus*	8,510	1.76		human gut	55,245	11.45	
	Enterococcus malodoratus	3,279	0.68		Klebsiella quasipneumoniae	7,614	1.57		Enterococcus gilvus	4,491	0.93	
Note:

* Mock microbial community in the positive control sample—Enterococcus avium, Staphylococcus aureus and Klebsiella pneumoniae.

Optimising sampling technique for collection of microbial keratitis on ex-vivo porcine model

To optimise the sampling technique on ex-vivo porcine model of microbial keratitis, comparisons were made between the different collection materials, both pre- and post-inoculation. Pre-inoculation, the average CT of 16S was not found to differ significantly between the collection materials (p = 0.909) (Table S3). Post-inoculation, the DNA concentration was found to differ significantly across the collection materials (p < 0.001) (Table S4; Fig. S3). The highest concentrations were observed for the Sugi® Eyespear and Isohelix™ SK-2S, with medians of 82.2 and 96.6 ng/ul respectively, whilst the lowest concentrations were observed for the PTFE 4mm, at a median of 0.19 ng/ul. For the average CT, no significant difference across the collection methods was detected for 16S (p = 0.242). The 16S: β-Actin ratio was consistent across all of the collection materials (p = 1.000).

Six out of the twelve collection methods provided sufficient DNA yield for 16S Nanopore sequencing and 16S copy number quantification—Sugi® Eyespear, Isohelix™ SK-2S, MW1021D Dryswab™ Polyester, MW130 Hospiswab™ Albumin, MW1041 Cotton and MW1021 Dryswab™ Rayon (Fig. 2). All three species of the mock microbial community inoculated onto the ex-vivo porcine eyes were detectable on nanopore sequencing. The Sugi® Eyespear swab was chosen for patient sample collection as it provided the highest mean rank DNA concentration that was sufficient for nanopore sequencing.

Figure 2 Comparison of swab collection yield, defined by relative abundances of mock community normalised by 16S copy number.

Proof-of-concept clinical study on microbial keratitis patient samples

The optimised sampling technique and bioinformatics pipeline were applied directly on two test patients with suspected microbial keratitis. Corneal, conjunctival and negative control swabs were collected from two consecutive patients presenting to the emergency department (Table 4; Table S5). Patient A had a majority of reads (497,995 out of total 613,482 reads, 81.2%) mapping to Serratia marcescens, which was in agreement with culture results. Patient B, who had been started on topical antibiotics prior to sampling, had no growth on culture after 5 days, but the majority of reads mapped to Bacillus subtilis (33,188 out of a total of 92,810 reads, 35.8%) on nanopore sequencing. Further Illumina Miseq 16S rRNA sequencing confirmed the presence of Bacillus in the affected eye of Patient B. Control samples from the unaffected contralateral conjunctival swabs and the respective negative control ‘air swabs’ had negligible reads (<1%) compared to the number of reads generated from the affected eye.

Table 4 Results of culture and 16S rDNA sequencing of patient samples aligned to NCBI 16S RefSeq Database at 100% similarity.

Sample	Total mapped reads of sample (Proportion of sample reads/total reads in sequencing run)	Bacterial taxa identified on nanopore sequencing (number of reads, relative abundance)	Culture results	Most abundant bacterial taxa identified on 16S V4 MiSeq (relative abundance)	
Patient A—total reads of sequencing run: 1,097,318	
Affected eye (Right cornea)	613,482 (55.9%)	Serratia marcescens (497,995, 81.2%);
Serratia nematodiphila (60,000, 9.78%);
Klebsiella aerogenes (9,247, 1.51%);
Kluyvera ascorbate (1,981, 0.32%);
Cutibacterium acnes (1,942, 0.32%)	Serratia marcescens	–	
Unaffected eye
(Left conjunctiva)	28 (0.0026%)	Serratia marcescens (6, 21.4%);
Streptococcus cristatus (3, 10.7%); Hungateiclostridium cellulolyticum (2, 7.14%);
Streptococcus parauberis (2, 7.14%);
Enterococcus avium (2, 7.14%)	–	–	
Negative control	26 (0.0024%)	Klebsiella pneumoniae (6, 23.1%);
Enterococcus avium (4, 15.4%);
Serratia marcescens (4, 15.4%);
Staphylococcus aureus (2, 7.69%);
Kluyvera ascorbate (1, 3.85%)	–	–	
Patient B—total reads of sequencing run: 273,987	
Affected eye (Left cornea)	92,810 (33.9%)	Bacillus subtilis (33,188, 35.8%); Staphylococcus caprae (5,875, 6.3%); Staphylococcus saccharolyticus (5,206, 5.61%);
Aggregatibacter segnis (4,109, 4.43%);
Cutibacterium acnes (2,998, 3.23%)	No growth in culture after 5 days	Bacillus (13.9%);
Dialister (8.6%);
Actinobacter (6.9%);
Rubrobacter (5.5%);
Staphylococcus (4.3%)	
Unaffected eye (Right conjunctiva)	1,301 (0.45%)	Snodgrassella alvi (935, 71.9%);
Escherichia fergusonii (48, 3.69%);
Anoxybacillus flavithermus (35, 2.69%);
Eikenella corrodens (24, 1.84%);
Thermicanus aegyptius (20, 1.54%)	–	–	
Negative control	6
(0.0022%)	Klebsiella pneumoniae (3, 50%);
Staphylococcus aureus (2, 33.3%);
Enterococcus avium (1, 16.7%)	–	–	

Discussion

We have developed a proof-of-concept study optimising the sample collection method and downstream bioinformatics pipeline for full-length 16S rRNA gene identification by nanopore sequencing in the setting of microbial keratitis.

The use of different swabs and collection methods for microbial keratitis has a direct effect on the DNA yield—the Sugi® Eyespear and Isohelix™ SK-2S swabs provided the highest DNA concentration. However, the ratio of host to microbial DNA recovery is similar across all collection methods. Differences in the DNA yield could be explained by the absorption efficacy of the swabs (Bruijns, Tiggelaar & Gardeniers, 2018). The absorption capacity of the swab materials is dependent upon the swab tip dimensions and morphology of the sorbent material—how tightly wound the sorbent material is to the shaft (Bruijns, Tiggelaar & Gardeniers, 2018). The Sugi® Eyespear swab, primarily designed for use in ophthalmic theatres due to its high tensile strength in absorbing fluids, has been proven to be effective in recovering DNA from corneal tissue in our study. Previous studies have also demonstrated that DNA recovery is inversely proportional to the fiber density on the swabs (Brownlow, Dagnall & Ames, 2012; Verdon, Mitchell & van Oorschot, 2014). Both Sugi® Eyespear and Isohelix™ SK-2S have swab tips made of cellulose fiber, which have high DNA-binding capacity (Su & Comeau, 1999).

Choice of primers affects the amplification efficiency of the 16S rRNA gene region, with primers that amplify the entire 16S rRNA gene spanning the V1-9 variable regions (27F and 1492R) providing better classification of reads compared to primers that only amplify a portion of the 16S rRNA region, as shown in studies by Winand et al. (2020) and Nygaard et al. (2020). Therefore, we have used the 27F and 1492R primer pairs provided in the commercially available ONT Rapid 16S barcoding kit. Other preliminary studies have also shown that sequencing the whole rrn operon (~4,300 bp), which includes the 16S rRNA gene, internal transcribed spacer (ITS) region, and the 23S rRNA gene, may provide better taxonomic resolution (Cuscó et al., 2019). However, as there is a lack of updated and curated rrn operon databases, users will need to retrieve and compile whole ribosomal operon reference database for their specific usage (Benítez-Páez & Sanz, 2017), whereas, curated or updated 16S rRNA reference databases are more readily available (RefSeq, 2020; Quast et al., 2013).

Another factor influencing the quality of nanopore sequencing is the choice of basecalling software (Rang, Kloosterman & De Ridder, 2018). We used the Guppy ‘flip-flop’ high accuracy model for basecalling, based on Wick et al’s study showing that the model, which utilises the recurrent neural network algorithm, performed better than the other basecalling programs (Albacore, Scrappie and Flappie) (Wick, Judd & Holt, 2019). Using Minimap2 (Li, 2018), we aligned our nanopore sequencing reads against the three different publicly available databases, NCBI 16S RefSeq, RDP and SILVA. Performance matrix comparison of thirteen different classification tools by Urban and colleagues revealed that Minimap2 provided robust alignments that were closely aligned to their mock community taxa (Urban et al., 2020). However, similar to the challenges encountered by Urban et al. (2020), we have had issues of high memory usage on Minimap2, which necessitated a reduction in the number bases loaded into memory to process in the query batch (command -K 100M). By comparing against a defined mock community, we observed differences in the taxonomic assignments between the databases—with the NCBI 16S RefSeq database clustered at 100% providing the most accurate assignments, which could be attributed to the differences in the database size and sequence validation steps (Balvočiūtė & Huson, 2017; Park & Won, 2018). The NCBI 16S RefSeq database is manually curated and near full length 16S sequences are preferentially selected (RefSeq, 2020). In our benchmarking steps, we have used the CD-HIT program (Li & Godzik, 2006), which employs heuristic greedy incremental clustering algorithm to cluster the reference sequences of the databases into 97%, 99% and 100% similarity to approximate taxonomic assignments. Schmidt, Matias Rodrigues & Von Mering (2015) assessed different clustering programs (hierarchical and heuristic algorithms) and observed that the CD-HIT program was robust, computationally efficient and provided reproducible clusterings. In our study, we have demonstrated that the 100% identity threshold provided a more optimal OTU assignment compared to 97% or 99% identity thresholds, which is consistent with previously published studies (Edgar, 2018; Mysara et al., 2017).

Nanopore sequencing reads were in concordance with current gold standard clinical microbiology culture techniques in Patient A. In the case of Patient B where cultures were negative, in the setting of previous antibiotic use, the nanopore sequencing result was in agreement with Illumina short-read sequencing suggesting an identification of a putative organism in the context of a false negative culture. The human ocular surface is paucibacterial. The conjunctival microbiome predominantly consists of Corynebacteria, Propionibacteria, Staphylococcus, and Streptococcus—with ‘approximately 1 bacterium for every 20 human conjunctival epithelial cell collected on conjunctival swab’ (Doan et al., 2016a). It is likely that the presence of Serratia marcescens and Bacillus subtilis at such high relative abundance in the patient samples, with correlating clinical signs and symptoms of infective keratitis, would constitute a positive test. A major challenge in applying high-throughput sequencing in clinical practice is distinguishing between true polymicrobial keratitis, ocular commensal or contaminant (Ung et al., 2020). Hence, we have taken meticulous steps to account for any potential contamination in our study—swabs of the unaffected contralateral eye and negative control swabs at the same time point and clinical environment of the patient had been taken, processed and sequenced in the same manner as the swabs of the affected eye. These negative control swabs had significantly fewer reads, less than 1% of the reads from the affected eye.

Another challenge in using full-length 16S rRNA sequencing is the difficulty in differentiating species and subspecies strains in certain bacterial genera with high sequence homology and similarity, notably the Bacillus subtilis species complex (Public Health England, 2018). This is illustrated in Patient B, whereby different taxonomic species within the Bacillus subtilis group such as Bacillus amyloliquefacians, Bacillus licheniformis, Bacillus velenzensis and Bacillus halotolerans have been assigned. Classical phenotypic tests by colonial appearance on culture, presence of ß-haemolysis, or biochemical tests to discriminate between these subspecies are unreliable. Other laboratory identification methods such as Matrix-Assisted Laser Desorption Ionisation–Time of Flight mass spectrometry to detect microbial protein composition is highly variable dependent upon endospore formation (Shu & Yang, 2017), whilst Multilocus Sequence Typing, which relies on PCR amplification and sequencing of six or seven well-conserved, housekeeping genes within the bacterial genome, is unable to provide distinct phylogenetic typing of Bacillus owing to the difficulty in designing primers for genetic sequences with high similarity. Although Pulse Field Gel Electrophoresis, which utilises endonuclease restriction enzymes, subsequent separation of DNA fragments by gel electrophoresis and staining under ultraviolet light for bands, is highly discriminatory, it is time consuming (over 30 h) and requires specialist laboratory equipment (Public Health England, 2018). Hence, Public Health England advocates initial clustering and genus identification by 16S rDNA for Bacillus, which could then be followed by more in-depth whole genome sequencing for more accurate strain characterisation (Public Health England, 2018).

Nanopore sequencing technology has the potential to provide rapid, real-time diagnosis of causative pathogens in a healthcare setting, with relatively low cost. Although in this proof of concept study, we have used 16S primers which specifically amplify bacterial DNA, non-biased deep metagenomic detection of pathogens and its antimicrobial resistance genomes from cultured clinical isolates using nanopore sequencing have previously been demonstrated (Schmidt et al., 2017; Szabó et al., 2016). Direct RNA sequencing on the nanopore platform would also enable identification of ‘live’ pathogens and host gene profiling for transcriptome signatures related to the infection (Lalitha et al., 2019). However, challenges still remain in terms of limiting background contamination (Glassing et al., 2016), reducing the error rates of sequencing and improving base-calling algorithms (Wick, Judd & Holt, 2019). Performing molecular diagnostics on ocular samples is inherently more difficult compared to other types of clinical samples, such as blood, urine or cerebrospinal fluid, as ocular samples are many magnitudes smaller in volume (Doan et al., 2016b) and are highly abundant in human cells (Lalitha et al., 2019).

This is the first study to evaluate the use of full-length 16S nanopore sequencing for detection of pathogens in microbial keratitis. We have optimised collection methods and demonstrated the bioinformatics pipeline for bacterial microbial keratitis. Our study is limited by the small sample size of patient cohort and the use of 16S rRNA primers, which specifically amplifies bacterial genome. To resolve this, larger clinical sample studies involving unbiased metagenomic sequencing are required to determine the sensitivity and specificity, as well as the cost effectiveness of nanopore sequencing.

Conclusion

We have optimised collection methods and demonstrate a novel workflow for identification of bacterial microbial keratitis using nanopore sequencing.

Supplemental Information

Supplemental Information 1 Pilot data comparing DNA yield of dry vs. pre-moistened swab at different time points.

Freshly enucleated porcine eyes (Sus scrofa domestica) were obtained as a by-product of the meat industry and transported to the laboratory under storage at 4 °C. Each eye was disinfected with Povidone-iodine 10% w/w for 1 min, followed by two rinses of sterile 0.9% Sodium Chloride for 1 min, and placed in an individual chamber of sterile 6-well culture plate (Sigma–Aldrich, Merck KGaA, Darmstadt, Germany). Using stereoscopic surgical loupes, a 4mm trephine punch (Acu-Punch®, Acuderm, Fort Lauderdale, USA) was used to create a single central anterior stromal corneal lesion (with debridement of the central 4mm). Each eye was then inoculated with 20 µL of 1 × 105 CFU/ml each of the mock community (Enterococcus avium, Staphylococcus aureus and Klebsiella pneumoniae). Negative control eyes were not inoculated with the mock community. The area was re-sampled with the respective swab conditions (dry swab vs. pre-moistened swab with sterile 0.9% Sodium Chloride) at two time points (30 min vs. 12 h) using Purflock® Ultra Standard (MWE Medical Wire, Corsham, UK) at room temperature, and placed immediately into a ZR BashingBead™ Lysis Tube containing 750 µl of DNA Shield™ (Zymo Research, Irvine, CA, USA) and stored at −80 °C until DNA extraction. DNA was extracted using ZymoBIOMICS DNA Miniprep kit (Zymo Research, Irvine, CA, USA) according to the manufacturer’s instructions. DNA concentration was determined fluorometrically using a Qubit dsDNA high-sensitivity assay (Thermo Fisher Scientific, Waltham, MA, USA).

Click here for additional data file.

Supplemental Information 2 Comparison of the 16S rRNA databases.

References: 1. 16S RefSeq records processing and curation. https://www.ncbi.nlm.nih.gov/refseq/targetedloci/16S_process/. Accessed February 17, 2020. 2. Cole J. R., Wang Q., Fish J. A., et al. Ribosomal Database Project: Data and tools for high throughput rRNA analysis. Nucleic Acids Res. 2014;42(D1):D633. doi:10.1093/nar/gkt1244 3. Quast C., Pruesse E., Yilmaz P., et al. The SILVA ribosomal RNA gene database project: Improved data processing and web-based tools. Nucleic Acids Res. 2013;41(D1):D590. doi:10.1093/nar/gks1219

Click here for additional data file.

Supplemental Information 3 Pre-inoculation average CT by collection material.

Data are reported as medians and ranges, with p-values from Kruskal–Wallis tests; bold p-values are significant at p < 0.05.

Click here for additional data file.

Supplemental Information 4 Post-inoculation average CT and DNA concentrations by collection material.

Data are reported as medians and ranges, with p-values from Kruskal–Wallis tests; bold p -values are significant at p < 0.05. *The ordering of collection materials, sorted from largest to smallest on the mean rank of the DNA concentration.

Click here for additional data file.

Supplemental Information 5 Comparing the effects of using NCBI 16S, RDP and SILVA databases at different levels of clustering (97%, 99% and 100% similarity) on patient samples.

Click here for additional data file.

Supplemental Information 6 Basic data statistics and QC.

Click here for additional data file.

Supplemental Information 7 Raw data for qPCR 16S Eubacteria copy number.

Click here for additional data file.

Supplemental Information 8 Raw data for qPCR 16S—swab comparison.

Click here for additional data file.

Supplemental Information 9 Raw data—Key for linking sample ID to swab type.

Click here for additional data file.

Supplemental Information 10 Raw data for qPCR Beta actin—swab comparison.

Click here for additional data file.

Supplemental Information 11 Bar graph comparing the effects of removal of unmapped, non-primary and supplementary reads on the taxonomic identification of the mock community.

Click here for additional data file.

Supplemental Information 12 Bioinformatics workflow.

Schematic diagram of bioinformatics workflow.

Click here for additional data file.

Supplemental Information 13 Post-inoculation DNA concentration by collection material.

Collection materials are sorted in order of the mean rank of DNA concentration.

Click here for additional data file.

West Midlands Collaborative Ophthalmology Network for Clinical Effectiveness & Research by Trainees (WM CONCERT) collaborators: Radhika Patel, George Moussa, Sreekanth Sreekantam, Pavitra Garala, Li Jiang, Sam Yuen Sum Lee, Ian de Silva, Aaron Ng, William Fusi-Rubiano, Yu Jeat Chong, Jesse Panthagani, Hedayat Javidi, Hemalatha Kolli, Kenan Damer, Xiao Li Chen, Usama Kanj, Richard Blanch.

Additional Information and Declarations

Competing Interests

Author Contributions

Human Ethics

Data Availability

Professor Nicholas J. Loman is an Academic Editor for PeerJ and has received Oxford Nanopore Technologies (ONT) reagents free of charge to support his research programme (but not for this study), travel expenses to speak at ONT events and an honorarium to speak at an ONT company meeting. All the swabs and collection methods were provided free of charge directly from the suppliers.

Nicholas J. Loman is a director of Microbial Genomics Ltd. Pablo Fuentes-Utrilla is an employee of Microbial Genomics Ltd.

Liying Low conceived and designed the experiments, performed the experiments, analyzed the data, prepared figures and/or tables, authored or reviewed drafts of the paper, and approved the final draft.

Pablo Fuentes-Utrilla conceived and designed the experiments, performed the experiments, authored or reviewed drafts of the paper, and approved the final draft.

James Hodson analyzed the data, prepared figures and/or tables, authored or reviewed drafts of the paper, and approved the final draft.

John D. O’Neil performed the experiments, authored or reviewed drafts of the paper, and approved the final draft.

Amanda E. Rossiter conceived and designed the experiments, performed the experiments, authored or reviewed drafts of the paper, and approved the final draft.

Ghazala Begum conceived and designed the experiments, performed the experiments, authored or reviewed drafts of the paper, and approved the final draft.

Kusy Suleiman performed the experiments, authored or reviewed drafts of the paper, and approved the final draft.

Philip I. Murray conceived and designed the experiments, authored or reviewed drafts of the paper, and approved the final draft.

Graham R. Wallace conceived and designed the experiments, authored or reviewed drafts of the paper, and approved the final draft.

Nicholas J. Loman conceived and designed the experiments, analyzed the data, prepared figures and/or tables, authored or reviewed drafts of the paper, and approved the final draft.

Saaeha Rauz conceived and designed the experiments, authored or reviewed drafts of the paper, and approved the final draft.

The following information was supplied relating to ethical approvals (i.e., approving body and any reference numbers):

The research followed the tenets of the Declaration of Helsinki and was approved by the Health Research Authority Ethics Committee (Rapid Diagnosis of Ocular Infections (RADAR); Reference: 11/EM/0274).

The following information was supplied regarding data availability:

Bioinformatics scripts and the DNA sequencing (FASTQ) files are available at European Nucleotide Archive (PRJEB37709): SAMEA7573840, SAMEA7573841, SAMEA7573842, SAMEA7573843, SAMEA7573844, SAMEA7573845, SAMEA7573846, SAMEA7573847, SAMEA7573848, SAMEA7573849, SAMEA7573850, SAMEA7573851, SAMEA7573852, ERX4706745, ERX4706746, ERX4706747, ERX4706748, ERX4706749, ERX4706750, ERX4706751, ERX4706752, ERX4706753, ERX4706754, ERX4706755, ERX4706756, ERR4836967, ERR4836968, ERR4836969, ERR4836970, ERR4836971, ERR4836972, ERR4836973, ERR4836974, ERR4836975, ERR4836976, ERR4836977, ERR4836978, SAMEA7573853, ERX4706757, ERR4836979, SAMEA7573854, ERX4706758, ERR4836980, SAMEA7556110, ERX4692670, ERR4822680.

The files are also available at figshare:

Low, Liying (2021): nanopore_mk_figshare.tar.gz. figshare. Dataset. DOI 10.6084/m9.figshare.13213898.v1.

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
