# Peer review of "Evaluation of full-length nanopore 16S sequencing for detection of pathogens in microbial keratitis"

_PeerJ, doi:10.7717/peerj.10778_

## Round 0.1 · original submission · Major Revisions

The reviewers have raised interest in this work; however, they have also raised substantial concerns about the paper, particularly in its general organization and discussion of biotechnological context. They have specifically commented that the paper needs more rigor in describing the methodology and clarification in the presentation of results.

Reviewer 1 ·

Basic reporting

In the present study, the authors present a novel molecular approach for detection of pathogens directly from clinical samples, using full length 16S sequencing on the Nanopore platform, in the context of the disease microbial keratitis. The authors investigated several different sample collection methods using an ex vivo porcine model of microbial keratitis to compare relative DNA recovery rates. They also developed a new bioinformatic analysis pipeline for taxonomic assignment. Finally, as a “proof-of-concept”, the authors applied their optimized pipeline and their most effective sample collection method to investigate the effectiveness of full-length nanopore 16S sequencing for pathogen detection from clinical samples. The need for developing newer, molecular approaches for culture-independent investigation of pathogens is immense, and the effort by these authors is commendable. Specifically, their efforts towards optimization of sample collection method seems thorough, and will be a great contribution to the field. However, the introduction and discussions do not help set up the research question very clearly or justify some of the method choices well, and some results need further clarification.

Experimental design

Firstly, the approach presented here is novel compared to standard molecular approaches in that it suggests the use of full length 16S reads, on the Nanopore sequencing platform. The authors however do not clarify why full length 16S reads is required for detection of pathogens in microbial keratitis. Therefore, for readers to better understand both the need and the adequacy of full length 16S sequencing, a discussion comparing the utility of variable region only (maybe V4 since its used here), full length 16S reads and whole genome-based approach for taxonomic profiling of common pathogens involved in microbial keratitis would be useful.
Secondly, the authors do not justify the use of Nanopore technique sufficiently, especially given that they mentioned the relatively high error rate of this platform (lines 118-119), which can be detrimental when discerning between closely related species.Therefore, a detailed discussion comparing full length 16S sequencing on the Nanpore and alternate platforms such as PacBio ccs platforms is essential, to provide required context for the present study. This is especially important because the authors claim that Nanopore sequencing has the potential for rapid, real-tie diagnosis with relatively low cost (lines 409-410), yet provide no information about the time, cost etc.
Finally, while the authors must be commended for including patient samples for validation of their method (and including negative controls), the sample size (n=2) is too small for adding any real value at this time, especially given that there is concordance in one patient and discordance for another, with respect to culture results. A “proof of concept” study might not require as many samples to provide sufficient power for statistical analysis, but as it stands the sample size of 2 is insufficient to draw any meaningful conclusion about the validity of the method. At least another culture positive sample should be included, and Illumina sequencing results for all 3 shown for comparison.

Validity of the findings

In Table 4, for patient B, what were the other species identified by the Illumina method? Did those species match with the Nanopore results, especially since B. subtilis constituted only 13.9% of the sample’s relative abundance in the Illumina results, indicating that it was likely not the most abundant species in that dataset. If the most abundant species is different in the two methods, this might be an issue, especially since the sample is culture negative. Showing top 5 species for each affected eye result would help understand the results better, given the low abundance of top species.
Lines 371-373: Reads as though nanopore results were in concordance with culture techniques for both patients, which is not true, please clarify.

Additional comments

How were the 3 species included in the mock community decided? The authors did not list E. avium as one of the common species found in bacterial keratitis (lines 77-81). Also, the major species detected in the clinical samples were S. marcescens and B. subtilis. Thus, the choice of species in the defined mix needs to be explained further. Including those species found in the clinical samples in the defined mix for bioinformatic pipeline optimization may help generate greater confidence in the method.
The data presented in Table 2 should be presented as bar charts (of relative abundance) for quick and easy comparison, with the counts included in a supplementary table possibly.
While the authors say that filtered reads better represented the mock communities (lines 278-286), details of how the filtering parameters were chosen, is not discussed, either in results or methods. To that end, a figure, maybe in supplementary, with visual representation of the bioinformatic processing steps would be valuable, with more details than in Figure 1.
Line 235-236: The authors claim that the bioinformatics scripts and fastq files are available at the GitHub repository (https://github.com/liyinglow/nanoporeMK), but I did not find any of those at that link (last cloned June 1st, 2020).
Supplementary Figure 1: Please replace with high resolution image for better readability.

Reviewer 2 ·

Basic reporting

This manuscript concerns the proof of concept application of Nanopore 16S amplicon sequencing as an alternative, quick, accessible, and inexpensive diagnosis tool for bacterial keratitis. The authors identify two important steps to explore and optimize 1) sampling instrument 2) bioinformatic pipeline for classification of the 16S data. The authors find that sampling tool has a big impact on the amount of DNA that is obtained. The authors also find that processing of mapping results and database clustering impacts the apparent classification of the data.
In general, I am a big fan of this type of proof of concept and I believe this type of application of Nanopore sequencing in a clinical setting is possible and extremely valuable. What the authors are doing is important work. However, using 16S for classification is not straight forward and is extremely contextual. On-top of this the high raw error rate complicates this further. I have tried to point out where in the paper I think this needs to be addressed.

1. A key misunderstanding is that 16S can be used for classification below genus level. There are many examples of different species with the same full length 16S sequences. Hence, you cannot in general do species level classification with 16S. Furthermore, in general it is accepted that two 16S sequences that are less than 97% identical usually originate from different species. The average read identity for Nanopore data is 95%, so it does not make sense to talk about species level classification in this respect either – your reads are almost always so different from the reference that they are considered a different species. So, on a fundamental theoretical level species level classification is not possible.

So why do the authors of this article as well as many others seem to observe decent species level classification of raw Nanopore 16S reads? We have had a bioinformatics PhD student investigating this for more than year and the factors seem to be many and these factors interact in complex ways. The primary factor is the reference database; if the reference database contains sequences from the bacteria in your sample, raw Nanopore reads tend to map to the correct references or closely related references (which can be different species!). So yes, you can observe correct species level assignments if the database you map to only or primarily contain sequences that match the species in your sample. But imagine that your sample contains a species not in the database, when you map to the database your reads will find the next best reference, which is a different species. In worse cases this can be different genus or even family! If this database only contains one closely related sequence most of the reads will map to this closely related sequence making it seem like a strong signal, even though it is wrong. The other factor is how to perform sensible mapping filtering. Nanopore data today has a median accuracy of around 95%, but the range is still 80 – 99%, where the low-quality reads are the real issue, as these reads tend to map more randomly, and hit many different species if you are unlucky.

Does it make sense to do 16S Nanopore sequencing then? It really depends on the use case and how the pitfalls above are addressed. In the case presented in this paper I think it might make sense. A way to do this would be for the authors must identify the species known to be relevant to keratitis and make sure these species are correctly represented in the databases. When doing analysis, only mappings to these species should be considered, as there is too much uncertainty about other mappings in general. The other mappings can be used qualitatively but not in a diagnostic setting before they have been validated. If a substantial amount of reads map to a specific genera known to cause keratitis this can be used as a crude indicator of infection and can likely be used as basis for early treatments if these treatments can be chosen based on only knowing the bacterial genera e.g. if you know all Bacillus infections can be treated a certain way. I think something like the might work, but it needs to be stated as a premise in the beginning with accompanying arguments and the manuscript needs to be purged of considerations of species level classification. It also needs to be backed up by empirical data, and here two patients simply isn’t enough.

2. In my opinion the introduction contains most of the relevant considerations for this topic, but the flow and argumentation is a bit messy and difficult to follow and many important points are first introduced in the discussion.

- Line 89: I am missing a transition from culture dependent to culture independent techniques and a very short mention of culture independent options and why sequencing is better than or a relevant alternative to qPCR (Only feasible to look for a couple of specific species with qPCR?).
- Line 89: I am also missing a sentence presenting the two sequence options: `metagenomic sequencing` and `targeted sequencing`.
- Line 89 – 94: Here you provide some really good examples of the strength of metagenomic sequencing, but I would like you to say it outright – A completely untargeted approach allows you to discover unexpected pathogens.
- Line 93: I a missing a bit more background information as to why the running cost is high, which you mention in line 116-117. I would state it here: pathogen DNA usually makes up << 0.1% and host DNA the rest (please find ref and check the numbers) of the sample DNA and hence requires deep sequencing only possible at specialized centers. Optional: You could add there are efforts pushing for solving this problem but they are not “production ready” yet and not compatible with a generic method that can be used everywhere (PMID: 31235920).
- Line 96: I would introduce the concept of marker genes and state that 16S rRNA is by far the most well established for bacteria, and hence the most obvious to use if you want a general overview of bacteria profiles in a sample. Also explain it is targeted by PCR which enables us to avoid background and sequence smaller parts of the genome, which drastically reduces sequencing depth requirements and therefor price.
- Line 98: The mention of OTUs in this sentence is weirdly specific. I think I would just say there are several examples of 16S being used to identify bacteria in clinical samples. Add more references than 1.
- Line 99: Here the authors appropriately include an overview of variables regions used for clinical studies (maybe add refs?), but I am missing some context here, which I think is key. Marker genes can be used but the resolution is generally lower than for `metagenomic` approaches (in some contexts the resolution goes down to genus level for 16S), which is a very important point to make here. The resolution is further reduced by use of only a fraction of the 16S gene. Use this space to qualify what primers you have used in your study. How much diversity do they cover or why this is not important as they cover the important genera for the disease (PMID: 22933715)? How big part of the 16S do they cover? etc. It would also be good to point out that databases has a huge impact on the `resolution` as well, which is why there are many specialized 16S databases out there (http://www.homd.org/).
- Line 104 -112: I like the points you are making here, but maybe add some stats to make the contrast to the aforementioned techniques/platforms more clear: How many 16S samples in how short time?
- Line 105 – 108: I would omit these sentences as these details are irrelevant to the points you are trying to make – Yes it can read 6 million bp reads, but your focus is sequencing 1500 bp.
- Line 114 – 119: I think I would move most of the examples up to when you introduce sequencing as a strategy, because they are universal problems irrespective of platform (ineffective sampling, low pathogen count/high host background and contamination, standardization) and where some are specific for the approach you are advocating (Nanopore error rate and how to handle it bioinformatically), which should stay here. I guess host DNA/low pathogen count is “solved” by using a targeted approach? Reagent contamination can be handled by using appropriate protocols, which you could mention, as your experiments does not address this specifically.
- Line 119: The key issue for using Nanopore for 16S sequencing is the error rate, which means conventional 16S pipelines cannot be used. There are several other papers (quick google shows > 20 papers e.g. PMID: 30815250 , PMID: 31980229 – find the best ones) that have dealt with this problem and you should cite and discuss their findings, so you can clearly show how your pipeline differs or if it is the same as some other pipelines give them credit and explain why you chose that particular pipeline.
- Line 120: I would state more directly in the first line that you focused on optimization of DNA extraction and bioinformatics pipeline to make 16S nanopore sequencing feasible.

3. Regarding sampling and DNA extraction I recommend using more `space` in the introduction discussing these steps as they are absolutely critical to the final results - as the authors do point out - and many researchers don’t consider this often enough. I would like to see some references to articles showing impact of sampling and DNA extraction to really underline this. This will nicely support your emphasis on the sampling experiment and support why you have chosen ZymoBIOMICS DNA Miniprep kit for DNA extraction, which uses the absolutely critical step of bead beating to obtain hard to lyse bacteria.

Experimental design

1. I am missing a clear definition of the success criteria for this study.
- Sampling: The criteria seems to be high DNA recovery. This can be misinterpreted, as most of this DNA will be host DNA anyway. Technically you are interested in higher 16S bacterial copy numbers. But I see what you mean and because you prove copy number is a function of DNA concentration, DNA concentration can be used as a proxy measurement.
- 16S sequencing: This is unclear. Is it simply to detect specific genera? What about false positives and false negatives? This is essential information to evaluate your results.

2. The patient cohort is too small. I know this is a proof of concept study, but even these can be too small. I would like to see at least 10 patients samples, and >20 would be better so you can estimate a crude success rate and convince to the community that this a valuable tool.

3. Line 157: How did you decide how many CFU to add to Ex vivo porcine model of microbial keratitis? Can you provide references that describe this model or support this amount with empirical data?
4. Line 164: I would like a better overview of the sampling scheme here. How many sample replicates were taken with each method. Was it 6? What does “sampling order was varied” mean? Hopefully, the same eye wasn’t sampled with multiple methods?
5. Line 183: Add a short line describing the purpose of these qPCRs.
6. Line 218: Some might wonder why you chose 27F-AGAGTTTGATCMTGGCTCAG; 1492R-CGGTTACCTTGTTACGACTT primers as they are not the best full length primers for capturing diversity (PMID: 22933715). Maybe add an explanation.
7. Line 240: How many patients?
8. Line 248: Can you provide a reference to a standard protocol for this “routine clinical practice”? My clinical knowledge is limited, but this might be different around the world.
9. Line 263: What protocol was used for 16S amplicon prep and sequencing? Is it the references by the end of the paragraph?

Validity of the findings

1. The bioinformatics pipeline is not novel and does not warrant the emphasis that it gets. The “filtering” performed on the mapping results is not filtering but rather basic clean up of mapping results – removing unmapped, secondary mappings and supplementary mappings is default in these types of workflows. Comparing “unfiltered” versus “filtered” is therefore not a relevant comparison. For filtering you would look at reference coverage (how much of the reference does my read cover) and reference/query identity (how different is my read from the reference). These are essential checks and cutoffs need to be qualified.

2. There are some major issues with the mock community validation. Several times there is talk of “species level assignments” (which is not possible with 16S), but the results from the mock data clearly shows that several other species and genera are targeted and the implications of this is for early diagnostics of keratitis is not addressed properly. Line 280-281: I would not expect this few reads to map with the mapping settings you used (only ~14% !!). This indicates that something is wrong. Maybe you need to do adapter trimming (e.g. porechop) for the mapping to work better, or the error rate is not what you think it is (estimate error rate for your data), or there is something else in your library that is not 16S or is concatenated 16S amplicons (check your read size distribution).

3. I am missing an explanation of why database clustering was considered as a strategy (Line 288). I actually think it is a good idea since it makes the mapping faster and species level assignments are not possible anyway and genera level assignments seem to be decent. But the reason presented is that it improves species level classification, which cannot be done in the first place. Clustering will lump similar sequences belonging to different species together, so of course clustering at lower identities will result in poorer “classification” – you risk removing your reference, which actually seems to be the case for Klebsiella pneumoniae in the SILVA database (Table 3). Clustering actually creates the problem I talk about in comment 1, where your data will map to the next best thing if your actual reference is not present in the database.

Line 276: Please provide us some basic data statistics and QC. Number of raw reads produced and an estimate of error rate (estimate from mapping data directly to the correct references and using cigars to calculate identity), aswell as a raw read length histogram.
Line 288-297: It is unclear what is going on here. Did you cluster the reference databases and then tried to classify them afterwards or are you comparing read classification with reference databases clustered at different identities? The last sentence ends with “… and 297 align them to the NCBI 16S RefSeq database at 100% similarity.” With a 5% error rate there is not going to be any reads that are 100% similar.
Line 312: I am not following this statement. From the 16S Ct values in supplementary table 4 the highest Ct is 23.7, which in my experience usually generates fine 16S amplicons and can be quantified with qPCR. Can you elaborate on the cut off you used?
Line 342: I don’t understand the meaning of “… Minimap2 provided robust alignments that were closely aligned to their mock community taxa”. As far as I can see from the Urban reference, minimap succeeded in mapping most reads from a mock community dataset. Does this mean it is better or is it simply because Minimap mapping settings are more lax?
Line 426: This line seems broken “primers amplifying genome for genus”

Additional comments

Line 102: Many people regularly sequence up to 500 bp amplicons on the MiSeq, so maybe use that number.

---

## Round 0.2 · accepted · Accept

Congratulations on comprehensively addressing the reviewers' concerns, and acceptance of your paper.

Reviewer 1 ·

Basic reporting

The authors have addressed most of the major concerns I had raised in my initial review, and I have do not have any major remaining concerns at this time, and I am happy to recommend this manuscript for publication.

Experimental design

N/A

Validity of the findings

N/A

Additional comments

N/A